# Hepatocyte-Specific *Fads1* Overexpression Attenuates Western Diet-Induced Metabolic Phenotypes in a Rat Model

**DOI:** 10.3390/ijms25094836

**Published:** 2024-04-29

**Authors:** Dushan T. Ghooray, Manman Xu, Hongxue Shi, Craig J. McClain, Ming Song

**Affiliations:** 1Department of Medicine, Division of Gastroenterology, Hepatology and Nutrition, University of Louisville School of Medicine, Louisville, KY 40202, USA; dushan.ghooray@louisville.edu (D.T.G.); m0xu0008@louisville.edu (M.X.); craig.mcclain@louisville.edu (C.J.M.); 2Department of Pharmacology and Toxicology, University of Louisville School of Medicine, Louisville, KY 40202, USA; hs3205@cumc.columbia.edu; 3Department of Medicine, Columbia University Irving Medical Center, New York, NY 10032, USA; 4Hepatobiology & Toxicology Center, University of Louisville School of Medicine, Louisville, KY 40202, USA; 5Alcohol Research Center, University of Louisville School of Medicine, Louisville, KY 40202, USA; 6Robley Rex Veterans Affairs Medical Center, Louisville, KY 40206, USA

**Keywords:** MASLD, fatty acid desaturase 1, lipidomics, insulin signaling, arachidonic acid, dihomo-γ-linolenic acid

## Abstract

Fatty acid desaturase 1 (FADS1) is a rate-limiting enzyme in long-chain polyunsaturated fatty acid (LCPUFA) synthesis. Reduced activity of FADS1 was observed in metabolic dysfunction-associated steatotic liver disease (MASLD). The aim of this study was to determine whether adeno-associated virus serotype 8 (AAV8) mediated hepatocyte-specific overexpression of *Fads1* (AAV8-Fads1) attenuates western diet-induced metabolic phenotypes in a rat model. Male weanling Sprague-Dawley rats were fed with a chow diet, or low-fat high-fructose (LFHFr) or high-fat high-fructose diet (HFHFr) ad libitum for 8 weeks. Metabolic phenotypes were evaluated at the endpoint. AAV8-Fads1 injection restored hepatic FADS1 protein levels in both LFHFr and HFHFr-fed rats. While AAV8-Fads1 injection led to improved glucose tolerance and insulin signaling in LFHFr-fed rats, it significantly reduced plasma triglyceride (by ~50%) and hepatic cholesterol levels (by ~25%) in HFHFr-fed rats. Hepatic lipidomics analysis showed that FADS1 activity was rescued by AAV8-FADS1 in HFHFr-fed rats, as shown by the restored arachidonic acid (AA)/dihomo-γ-linolenic acid (DGLA) ratio, and that was associated with reduced monounsaturated fatty acid (MUFA). Our data suggest that the beneficial role of AAV8-Fads1 is likely mediated by the inhibition of fatty acid re-esterification. FADS1 is a promising therapeutic target for MASLD in a diet-dependent manner.

## 1. Introduction

MASLD is currently the most common liver disease worldwide and affects approximately 30% of the population [1]. A small portion of persons with MASLD will progress to nonalcoholic steatohepatitis, now termed “metabolic dysfunction-associated steatohepatitis (MASH) [2]”, cirrhosis, and hepatocellular carcinoma (HCC) [3,4]. The mechanisms underlying MASLD development and progression are complex and remain elusive [5,6].

Dysregulated lipid metabolism is a key feature of MASLD, characterized by altered hepatic lipid composition and transcriptomics. Accumulated evidence from human studies has shown that long-chain polyunsaturated fatty acids (LCPUFAs), including arachidonic acid (AA, 20:4n-6), eicosapentaenoic acid (EPA, 20:5n-3), and docosahexaenoic acid (DHA, 22:6n-3) in hepatic lipid pools were significantly decreased in MASH patients compared to healthy controls and/or patients with simple steatosis [7,8]. Of note, the essential fatty acid (EFA) precursors of LCPUFA, including linoleic acid (LA, 18:2n-6) and α-linolenic acid (ALA, 18:3n-3) were unaltered [7], suggesting that decreased AA, EPA, and DHA were not due to EFA deficiency, likely owing to dysregulated conversion from essential fatty acids to LCPUFA.

FADS1 and FADS2 are membrane-bound desaturases that catalyze the synthesis of LCPUFAs [9]. LCPUFAs are biologically active components of membrane phospholipids that regulate hepatic gene transcription through the modulation of transcription factors in a ligand-dependent manner [10]. Genome-wide association studies (GWASs) have shown that single-nucleotide polymorphisms (SNPs) in the *FADS* gene cluster (*FADS1-2-3*) are associated with several metabolic traits [11,12]. In line with this, hepatic FADS1 SNP and decreased hepatic FADS1 activity were observed in MASLD and MASH patients [13,14,15]. However, the biological function studies from FADS1 deficient mice exhibit mixed results [16,17]. While Powell and colleagues showed that FADS1 deficiency is beneficial in improvements in obesity, diabetes, and atherosclerotic cardiovascular disease [17], another study demonstrated FADS1 deficiency worsens high-fat diet induced hepatic steatosis [16]. Further, antisense oligonucleotides (ASOs)-induced loss of FADS1 activity in low-density lipoprotein receptor (LDLR) null mice promotes hepatic inflammation, hypercholesterolemia, and atherosclerosis, and increases hepatic free cholesterol and cholesterol ester levels, but reduces hepatic triglyceride in an n3 substrate fatty acid-enriched diet [18].

Collectively, these results suggest that FADS1 function is diet- and context-dependent. Given that FADS1 activity is reduced in MASLD patients [13,14] and FADS1 deficiency worsens hepatic steatosis and cholesterol metabolism [15,18], we hypothesize that overexpression of *Fads1* in hepatocytes improves MASLD and metabolic phenotypes. In this study, we demonstrated that hepatocyte specific overexpression of *Fads1* using AAV8 is beneficial in the improvement in metabolic phenotypes in a rodent model fed a western diet. Our study provides evidence that FADS1 is a viable target in the preclinical models of metabolic disorders.

## 2. Results

### 2.1. Hepatocyte-Specific Fads1 Overexpression Significantly Attenuates the Metabolic Phenotypes Induced by High Fructose-Containing Diets

Although the body weight was not increased in rats fed with either the LFHFr or HFHFr diet, the white adipose tissue weight (eWAT) as well as eWAT/body weight ratio were significantly increased in HFHFr-fed rats compared to chow-fed rats (healthy control, without AAV injection). Hepatocyte-specific *Fads1* overexpression with AAV8-Fads1 led to significantly decreased body weight, liver weight, liver/body weight ratio, epididymal fat weight and epididymal fat weight/body weight ratio in either LFHFr or HFHFr-fed rats compared to the controls, as shown in Figure 1. Both LFHFr and HFHFr-fed rats exhibited smaller cecum as well as decreased cecum/body weight ratio compared to chow-fed rats, and that was not altered by AAV8-Fads1. AAV8-Fads1 treatment led to a decreasing trend in the energy intake in rats fed with the LFHFr diet compared to AAV8-Blank controls. The energy efficiency ratio was significantly reduced in AAV8-Fads1-treated rats compared to chow controls. Although fasting blood glucose levels showed a trend toward an increase in both LFHFr and HFHFr-fed rats compared to chow-fed rats, that was blunted by AAV8-Fads1. HOMA-IR and fasting blood glucose levels exhibited a similar trend of alterations. AAV8-Fads1 did not alter the plasma ALT levels, and it led to a decrease in plasma AST levels in HFHFr-fed rats, but that was not statistically significant (Table 1).

### 2.2. Hepatocyte-Specific Fads1 Overexpression Decreases Hepatic Cholesterol and Plasma Triglyceride Levels

To evaluate the effects of hepatocyte *Fads1* overexpression on hepatic fat accumulation and blood lipids levels, we measured liver total triglyceride and cholesterol levels. HFHFr feeding led to a significant increase in hepatic triglyceride (approximately 3-fold) and cholesterol (approximately 2-fold) levels compared to chow-fed rats. While AAV8-Fads1 markedly decreased hepatic cholesterol level in HFHFr-fed rats, it did not decrease hepatic triglyceride level. Although LFHFr feeding led to a mild increase in hepatic triglyceride and cholesterol contents, the differences did not reach statistical significance. Moreover, AAV8-Fads1 did not alter hepatic lipid content in LFHFr-fed rats. Liver histology exhibited similar findings as biochemical measurements (Figure 2A,B).

While AAV8-Fads1 markedly decreased plasma triglyceride levels in HFHFr-fed rats, it did not lead to a significant change in LFHFr-fed rats compared to the AAV8-blank controls. Moreover, AAV8-Fads1 significantly increased plasma NEFA levels in both LFHFr and HFHFr-fed rats compared to the AAV8-blank controls. However, plasma cholesterol level was not altered by AAV8-Fads1 in either LFHFr or HFHFr-fed rats compared to the AAV8-blank controls. Collectively, these data suggest that hepatocyte *Fads1* overexpression differentially altered hepatic and plasma lipid profiles (Figure 2C).

### 2.3. Hepatocyte-Specific Fads1 Overexpression Improves Glucose Tolerance and Insulin Signaling

To determine the effect of hepatocyte Fads1 overexpression on glucose metabolism, we performed GTT and ITT on rats exposed to the experimental diet for 5 weeks and 6 weeks, respectively. AAV8-Fads1 led to significantly improved glucose tolerance in LFHFr-fed rats but was not significantly improved in HFHFr-fed rats, as shown by GTT (Figure 3A). AAV8-Fads1 did not significantly alter insulin action in either LFHFr or HFHFr-fed rats compared to the AAV8-blank controls, as shown by ITT (Figure 3B). Collectively, these data suggest that hepatocyte *Fads1* overexpression enhanced insulin secretion but did not alter the whole-body insulin action [19].

To understand the effects of hepatocyte *Fads1* overexpression on organ-specific insulin signaling, we further assessed insulin signaling in liver, skeletal muscle, and adipose tissue by Western blot. Our data show that insulin induced Akt phosphorylation was not markedly altered in the liver as a result of different diets and AAV8 treatment. However, insulin induced Akt phosphorylation in skeletal muscle was improved by AAV8-Fads1 in LFHFr-fed rats. HFHFr feeding led to significantly reduced insulin stimulated Akt phosphorylation in adipose tissue compared to LFHFr-fed rats, and that was not altered by AAV8-Fads1. Insulin induced GSK-3β phosphorylation was not evident in the liver and adipose tissue as a result of different diets and AAV8 treatment. However, insulin-induced GSK-3β phosphorylation in the muscle was blunted by either LFHFr or HFHFr feeding, and that was not significantly improved by AAV8-Fads1 (Figure 4). Collectively, these data suggest that improved skeletal muscle insulin signaling by hepatocyte *Fads1* overexpression likely contributed to the improved glucose tolerance in LFHFr-fed rats but did not alter the whole-body insulin action.

### 2.4. Hepatocyte-Specific Fads1 Overexpression Increases Hepatic FADS1 Level and Functional Activity in LFHFr and/or HFHFr-Fed Rats

To verify that AAV8 was successfully delivered to the liver, we performed a quantitative real-time PCR to detect AAV-specific sequences and inverted terminal repeats (ITRs) from extracted liver DNA. ITRs are the only conserved sequences of viral origin in recombinant AAVs and are essential for genome packaging [20,21,22]. Our data show robust increased ITR expression in the livers of rats with AAV8 transfection compared to those without AAV8 transfection (chow-fed rats). Moreover, the DNA copy numbers are similar in the rats with AAV8 transfection, indicating an equal dose of AAV8 injection (Figure 5A). However, hepatic *Fads1* expression was significantly increased in the rats with AAV8-Fads1 compared to those with AAV8-Blank (Figure 5B). To validate the effect of AAV8-Fads1 injection on FADS1 activity, we performed hepatic lipidomics analysis. The AA(C20:4n6)/DGLA(C20:3n6) ratio is an estimation of FADS1 activity [14]. Our data showed that hepatic FADS1 activity (in terms of the AA/DGLA ratio) was markedly reduced by HFHFr compared to the chow-fed rats in the phospholipid (PL) and total FA pools, and that it was rescued by AAV8-Fads1. However, this effect was not evident in LFHFr-fed rats (Figure 5C), suggesting that AAV8-mediated *Fads1* overexpression increased FADS1 activity in a diet-dependent manner. Moreover, the effect of AAV8-Fads1 treatment on the n-6/n-3 ratio was not significant. Enhanced hepatocyte FADS1 expression was further shown through immunohistochemical staining in the liver section (Figure 5D).

### 2.5. Effects of Hepatocyte-Specific Fads1 Overexpression on the Alterations of Hepatic Fatty Acids Composition in LFHFr and HFHFr-Fed Rats

LFHFr feeding led to a significant increase in hepatic total fatty acids level (27%) in the free fatty acid (FFA) pool compared to the chow-fed rats, and it was further increased to 43% by AAV8-Fads1; this result was not observed in HFHFr-fed rats. Although hepatic total fatty acids level in the diacylglycerol (DAG) pool was not altered by either LFHFr or HFHFr diet, it was markedly decreased by AAV8-Fads1 in LFHFr-fed rats. HFHFr feeding led to a marked increase in total fatty acids level (4.5-fold) in the class of cholesterol ester (CE) compared to the chow-fed rats, and to a less extent (2.5-fold) by AAV8-Fads1. Similar alterations, but to a lesser degree, were observed in the triacylglycerol (TAG) pool. Hepatic total fatty acids levels in the class of phospholipid (PL) were markedly reduced by either LFHFr or HFHFr compared to the chow-fed rats and were not altered by AAV8-Fads1. Of note, hepatic total cholesterol level was markedly increased by HFHFr feeding compared to chow-fed rats, and that was attenuated by AAV8-Fads1 (Figure 6A). Next, we analyzed the total saturated fatty acid (SFA), monounsaturated fatty acid (MUFA) and PUFA levels in different classes of lipid. Hepatic total SFA in the CE and TAG pool were markedly increased by HFHFr feeding compared to the chow-fed rats, and that was decreased by AAV8-Fads1 in the CE pool. However, hepatic total SFA in the PL was significantly decreased by both LFHFr and HFHFr feeding compared to the chow-fed rats, and no significant alterations were induced by AAV8-Fads1 across the different lipid pools except for CE (Figure 6B). Hepatic total MUFA levels were robustly increased by LFHFr and/or HFHFr feeding across all the classes of lipid pools. Of note, the increase in hepatic total MUFA by LFHFr feeding was significantly blunted by AAV8-Fads1 in the pools of DAG and PL, and that was not observed in HFHFr-fed rats. While the increased MUFA expression by HFHFr in the class of CE was markedly reduced by AAV8-Fads1, no significant changes were observed by LFHFr. Moreover, hepatic total PUFA levels in the pools of FFA, DAG, TAG and PL were markedly decreased in rats fed with both LFHFr and HFHFr diets, and that was not significantly altered by AAV8-Fads1. However, hepatic total PUFA, as well as SFA and MUFA in the class of CE were all markedly decreased by AAV8-Fads1 in HFHFr-fed rats, and that was consistent with the alterations in the total cholesterol and CE (Figure 6B). Thus, hepatocyte Fads1 plays a significant role in the regulation of hepatic lipid homeostasis with a more pronounced effect on the CE class of lipids.

We further analyzed the effects of hepatocyte *Fads1* overexpression on the alterations to the composition and concentration of individual fatty acids (Figure 6C and Appendix A). C16:0 in the DAG and TAG pools was significantly increased by LFHFr and further increased by AAV8-Fads1. Since the majority of dietary essential fatty acids are catabolized by β-oxidation [23] and the released carbons are recycled into newly synthesized 16:0 [24], we speculate that the increased C16:0 may be due to the increased PUFA metabolism by AAV8-Fads1. C18:0 in PL pool was significantly decreased by LFHFr and rescued by AAV8-Fads1. C18:0 in the DAG pool was significantly increased by AAV8-Fads1 in HFHFr-fed rats. Of note, C16:1n7 and C18:1n7 were robustly increased across all the lipid pools by LFHFr feeding, and to a less extent by HFHFr. C18:1n7 in the TAG and PL pools, and C16:1n7 in the DAG pool were significantly decreased by AAV8-Fads1 in LFHFr-fed rats, suggesting suppressed C16:0 desaturation. C18:1n9, the major component of MUFAs, was significantly increased by LFHFr across all the lipid pools, and to a great extent by HFHFr; this result was significantly decreased by AAV8-Fads1 in HFHFr-fed rats, suggesting suppressed C18:0 desaturation. Collectively, these data suggest that hepatocyte *Fads1* overexpression likely inhibited hepatic lipogenesis through suppressed desaturation by SCD1. Given that *Scd1* is a target gene of SREBP-1, and SREBP-1 was inhibited by LCPUFAs [10,25], this led to the hypothesis that hepatocyte *Fads1* overexpression inhibits lipogenesis through enhanced LCPUFAs synthesis. In support of this, AAV8-Fads1 led to decreased hepatic 18:2n6 in LFHFr rats and 20:3n6 in HFHFr-fed rats. Moreover, AAV8-Fads1 also led to increased 20:4n6 in FFA and DAG pools and increased 22:6n3 in FFAs in LFHFr-fed rats (Figure 6C).

## 3. Discussion

The principal finding of this study is that hepatocyte *Fads1* overexpression by AAV8 significantly increased hepatic FADS1 activity and ameliorated metabolic phenotypes in rats fed with either LFHFr or HFHFr diets, including decreased body, liver, and white adipose tissue weight (Figure 1), and was associated with improved glucose tolerance (Figure 3A), hepatic lipid accumulation, and plasma triglyceride level (Figure 2). It appears that hepatocyte *Fads1* overexpression plays a beneficial role in the improvement in metabolic phenotypes in a context-dependent manner. Of note, hepatocyte *Fads1* overexpression differentially alters the metabolic phenotypes induced by either LFHFr or HFHFr, as characterized by markedly reduced hepatic total cholesterol as well as cholesterol ester levels in HFHFr-fed rats (Figure 6), and improved glucose tolerance in LFHFr-fed rats.

Of note is the finding that hepatic CE, but not the free cholesterol, was markedly increased by HFHFr feeding, and this was blunted by AAV8-Fads1, suggesting that cholesterol storage was decreased by *Fads1* overexpression [26]. Interestingly, Gromovsky et al.’s study showed that *Fads1* knockdown exacerbates hypercholesterolemia and increases free cholesterol level in the livers of LDLR null mice, likely through LXR-dependent and independent mechanisms [18], highlighting a critical role of FADS1 in the regulation of cholesterol metabolism. The mechanisms underlying *Fads1* overexpression induced reduction in hepatic cholesterol storage remain elusive. Our data show that plasma cholesterol levels were not altered by hepatocyte *Fads1* overexpression, suggesting that cholesterol uptake and release to the plasma were unlikely to be affected. Therefore, we speculate that reduced hepatic CE by hepatocyte *Fads1* overexpression is likely due to reduced cholesterol synthesis and/or increased excretion to intestine [26]. A previous study showed that AA-derived metabolites, leukotrienes (LTs) and lipoxins (LXs), promote reverse cholesterol transport (RCT) through upregulation of hepatic bile salt export pump *Abcb11* expression [27]. In line with this, the gene expression involved in bile acids synthesis and excretion, including *Cyp7a1*, *Abcg5* and *Abcg8*, was markedly upregulated through increased endogenously synthesized n-3PUFAs in a high fat diet-induced MASLD murine model [28], suggesting that n-3PUFAs may facilitate cholesterol excretion. Future study on whether Fads1 regulates hepatic cholesterol ester level via RCT is warranted.

In fact, a role for Fads1 in cholesterol metabolism was also supported by multiple GWASs. First, it has been shown that genetic variants in the FADS gene cluster (FADS1-2-3) were associated with plasma concentrations of HDL and LDL-cholesterol levels [29,30]. Second, the genes encoding the enzymes of LCPUFA metabolism, *ALOX5*, *ALOX12*, and *ALOX15* gene loci, were associated with alterations in plasma cholesterol levels. Moreover, robust association was found in the locus containing *ALOX5* [27]. Interestingly, a SNP, rs174537 near FADS1, which is the most strongly associated with AA level, was also associated with synthesis of ALOX5 products, but not COX products [31]. Whether or not and how FADS1 dictates AA-derived metabolites’ entry to ALOX pathways remains elusive.

Despite significantly reduced hepatic CE, hepatic triglyceride levels were not apparently decreased by hepatocyte Fads1 overexpression. Nonetheless, hepatic MUFA levels were significantly reduced in several classes of lipids, suggesting that FADS1 may inhibit lipogenesis via inhibition of MUFA re-esterification [32]. Moreover, fasting plasma triglyceride levels were markedly decreased by AAV8-Fads1 in HFHFr-fed rats. It is known that LCPUFAs are strong inhibitors of SREBP-1c. LCPUFA deficiency leads to the activation of SREBP-1c, increased triglyceride synthesis and VLDL secretion, and subsequent hypertriglyceridemia. On the other hand, endogenously synthesized LCPUFAs from essential fatty acids or dietary supplementation with PUFAs, such as ARA and DHA, suppressed SREBP-1c activation and normalized plasma triglyceride level [25,33]. Our data showed that hepatocyte Fads1 overexpression significantly increased hepatic AA and DHA levels in FFA pools, suggesting that increased endogenous LCPUFA synthesis through enhanced FADS1 activity is a likely mechanism contributing to the decreased plasma triglyceride levels in HFHFr-fed rats.

In contrast to its plasma triglyceride lowering effects, hepatocyte *Fads1* overexpression results in elevated fasting plasma NEFA (Figure 2C). The NEFAs in the plasma arise mostly from hydrolysis of triglycerides in adipose tissue in the fasting state [34]. In line with this, reduced white adipose tissue weight was observed in HFHFr-fed rats treated with AAV8-Fads1 compared to the AAV8-blank controls. Whether or not and how hepatocyte *Fads1* overexpression led to increased adipose tissue lipolysis remains elusive. Although an elevated fasting NEFA level is thought to promote insulin resistance, this paradigm is not always true [34]. In the current study, AAV8-Fads1 led to improved insulin signaling and glucose tolerance despite elevated fasting plasma NEFAs. Since hepatocyte *Fads1* overexpression rats gain less body weight relative to their energy intake as shown by reduced energy efficiency ratio, this suggests increased energy expenditure. We postulate that hepatocyte *Fads1* overexpression may increase β-oxidation, which consumes more energy. Future studies are needed to test this hypothesis.

An interesting finding is that hepatocyte *Fads1* overexpression significantly improved glucose tolerance in LFHFr-fed rats, as shown by improved GTT but not ITT, suggesting that increased insulin secretion but not insulin action contributes to the improved glucose tolerance [19]. In fact, GWASs have repeatedly shown that SNPs in the *FADS* gene cluster were associated with glucose metabolism [35,36,37]. Our study provides evidence that enhancing FADS1 activity in hepatocytes, which mimics gain-of-function of *FADS1*, effectively improves glucose tolerance likely through augmented insulin secretion. Our lipidomics data showed that AAV8-Fads1 significantly increased hepatic AA and DHA levels in the class of FFA. Given that synthesized LCPUFA by the liver is secreted into the bloodstream [24,38], we posit that the improved glucose tolerance may be attributed to increased β-cell insulin secretion promoted by LCPUFA [39]. In supporting this, a human study showed that the higher FADS1 activity was associated with a lower risk of type 2 diabetes [40].

Of note, FADS1 activity determines the lipid mediator balance in a highly diet-specific manner, since omega-6 (n-6) and omega-3 (n-3) LCPUFAs compete for the same desaturase and elongase enzymes during the synthesis of AA, EPA and DHA [41,42]. A study by Gromovsky et al. showed that *Fads1* knockdown led to enhanced hepatic inflammatory response characterized by increased gene expression of macrophage markers. In contrast, hepatic triglyceride content was decreased and associated with the downregulation of lipogenic gene expression by *Fads1* knockdown in the low-density lipoprotein receptor (LDLR)-null mice, particularly in animals fed the n-3 LCPUFA substrate (ETA) enriched diet. *Fads1* knockdown promotes macrophage phenotype shift to classic M1 activation and pronounced systemic inflammation in response to LPS stimulation compared to controls, likely owing to the reduced secretion of pro-resolving lipid mediators from macrophages [18]. Interestingly, the study of Athinarayanan et al. showed increased hepatic fat accumulation in *Fads1* null mice when exposed to a high-fat diet [16]. Of note, these null mice were supplemented with 0.6% AA in the diet since they cannot survive more 12 weeks without dietary AA supplementation [43]. Consistent with this, the study showed that gain-and-loss of *FADS1* function by genetic knockdown and re-expression led to a more pronounced effect on n-6 LCPUFAs, and to a lesser extent for the n-3 LCPUFAs. Reintroduction of the FADS1 activity was not able to increase the levels of omega-3 LCPUFAs [16], possibly owing to the AA supplementation. In line with this, both FADS1 and FADS2 activity were increased with a more pronounced effect on the n-3 pathway when endogenously or exogenously supplemented with n-3 LCPUFAs in the high fat diet [44]. Collectively, these studies underscored a diet-dependent manner of FADS1 action, a caveat when considering FADS1 as a therapeutic target.

There are several limitations of the current study. The dose of AAV8 was adopted from previously published work. We expect that optimization of AAV8 dose through titration would lead to more pronounced effects on the improvement in metabolic phenotypes. Another limitation is that we did not assess the diet-specific effect of *Fads1* overexpression by using special n-3 and n-6 enriched diets. A previous study revealed that the bottom line of FADS targeted therapy is the optimization of n-6/n-3 ratio or restoration of hepatic n-3 LCPUFA contents [44].

In summary, our study provides evidence that hepatocyte FADS1 activation, which mimics gain-of-function of *Fads1* variants, is beneficial in the improvement in metabolic phenotypes in a context dependent manner. Future study with pharmacological activation of FADS1 in a MASLD animal model with n3 and/or n6 enriched LCPUFA diets will pave the way to the translation of FADS1 as a therapeutic target.

## 4. Materials and Methods

### 4.1. Animal Experiments

Male weanling Sprague-Dawley rats [45] (40 g±) were purchased from the Harlan Laboratories (Indianapolis, IN, USA). On arrival, the animals were group housed in a temperature- and humidity-controlled room with a 12:12 h light-dark cycle and were acclimated on a standard pelleted rodent chow diet. Hepatocyte-specific *Fads1* overexpression was achieved via the tail vein injection of recombinant AAV8 vector containing an albumin promoter with either rat *Fads1* sequence (AAV8-Fads1) at the dose of 1.5 × 10^11^/rat one week before the experimental diet [46,47] or AAV8 vector containing only albumin promoter (AAV8-Blank), which served as control (ABM Inc., Richmond, BC, Canada). Rats were then randomly assigned to five groups: chow diet without AAV, low-fat high-fructose diet with AAV8 blank control (LFHFr-AAV8-Blank), LFHFr with AAV8-Fads1 (LFHFr-AAV8-Fads1), high-fat high-fructose diet with AAV8 blank control (HFHFr-AAV8-Blank), and HFHFr diet with AAV8-Fads1 (HFHF-AAV8-Fads1) (n = 8 rats/group), respectively, for 8 weeks. Ten percent or 60% of calories were derived from the fat (beef tallow) in LFHFr diet (20 kcal% protein, 70% kcal% carbohydrate, 10% kcal% fat, custom diet D18080701, Research Diets, Inc., New Brunswick, NJ) or HFHFr diet (20 kcal% protein, 20 kcal% carbohydrate, 60 kcal% fat, custom diet D14031902, Research Diets, Inc., New Brunswick, NJ, USA), respectively. The regular rodent chow diet (5010, LabDiet, St. Louis, MO, USA) served as a control for the refined diet. Tap water or tap water containing 30% fructose (*w*/*v*) (Millipore Sigma, St. Louis, MO, USA) was given ad lib during the entire experiment. Fructose enriched drinking water was changed twice a week. At the end of the experiment, all the animals were sacrificed under anesthesia with ketamine/xylazine (100/10 mg/kg I.P. injection) after overnight fasting. To measure insulin signaling in vivo, rats received a bolus injection of 1 unit/kg insulin or saline through the portal vein after overnight fasting [48,49]. The liver, gastrocnemius muscle, and epididymal adipose tissues were harvested 5 min after the injection. One milliliter of blood was collected from the inferior vena cava before insulin injection and stored in citrated plasma at −80 °C for further analysis. Portions of the liver, gastrocnemius muscle, intestine, and epididymal adipose tissue were fixed with 10% formalin for subsequent sectioning, while other portions were snap-frozen with liquid nitrogen. All studies were approved by the University of Louisville Institutional Animal Care and Use Committee, which is certified by the American Association of Accreditation of Laboratory Animal Care.

### 4.2. Liver Enzyme and Plasma Biochemical Assays

Liver enzymes and plasma biochemical assays were performed with commercially available kits according to the manufacturer’s instructions [45]: alanine aminotransferase (ALT), aspartate aminotransferase (AST), glucose, cholesterol, triglyceride (Infinity, Thermo Electron, Melbourne, Australia); non-esterified fatty acids (NEFA) (Wako Chemicals, Richmond, VA, USA); insulin (Lino Research, St. Charles, MO, USA).

### 4.3. Detection of AAV DNA in the Liver by Quantitative Real-Time PCR

DNA was extracted from approximately 20 mg of rat liver tissue using the QIAamp DNA Mini Kit (Qiagen, Germantown, MD, USA) according to the manufacturer’s instructions. Quantitative PCR (qPCR) was performed on a StepOne Real-Time PCR System (Applied Biosystems, Carlsbad, CA, USA) using primers for inverted terminal repeats (ITRs), which is the only conserved sequences of viral origin in recombinant AAVs. (ITR2 forward 5′-ggaacccctagtgatggagtt-3′, reverse 5′-cggcctcagtgagcga-3′) [21]. Hepatic AAV DNA levels were normalized to host β-actin.

### 4.4. Hepaticer FADS1 Assay by ELISA

Rat liver FADS1 was determined using ELISA kit (ELK Biotechnology Co., Ltd., Denver, CO, USA) with high specificity and sensitivity without significant cross-reactivity according to the manufacturer’s instruction.

### 4.5. Histology and Immunohistochemistry

Formalin-fixed, paraffin-embedded liver sections were cut at 5 µm thickness and stained with hematoxylin and eosin (H&E). For immunohistochemistry, formalin-fixed and paraffin-embedded liver sections (5 µm) were deparaffinized and rehydrated, followed by antigen retrieval and blocking, then incubated with primary antibody against FADS1 (1:100 dilution, Cat#: MA5-42500, Invitrogen, Waltham, MA, USA) overnight at 4 °C. The negative control was performed with PBS without primary antibody. Subsequently, the sections were incubated with horseradish peroxidase-conjugated secondary antibody for 1 h at room temperature, then stained with DAB (Cat#: SK-4105, Vector Laboratories Inc., Newark, CA, USA) for 5 min followed by counterstaining with hematoxylin, dehydration and mounting.

### 4.6. Hepatic Triglyceride Assay

Liver tissues were homogenized in ice-cold phosphate buffered saline. Hepatic total lipids were extracted with chloroform/methanol (2:1) according to the method described by Bligh and Dyer [50]. Hepatic triglyceride and cholesterol content were determined using commercially available reagents (Infinity, Thermo Electron, Melbourne, Australia) according to the manufacturer’s instructions.

### 4.7. Glucose Tolerance Test (GTT)

At week 5, rats were I.P.-injected with 50% glucose (Millipore Sigma, St. Louis, MO, USA) at 1 g/kg body weight after 16 h of fasting. Glucose levels were obtained at 0, 15, 30, 60, and 120 min. Blood samples were obtained by tail nick and glucose levels were measured using an ACCU-CHEK Aviva glucose meter (Roche Diagnostics, Basel, Switzerland) [51,52].

### 4.8. Insulin Tolerance Test (ITT)

At week 6, rats were I.P.-injected with recombinant human insulin (Humulin R, Lilly USA, LLC, Indianapolis, IN, USA) at the dose of 0.50 U/kg body weight after 5 h of fasting [51,52]. Glucose levels were measured at time points 0, 15, 30, 60, and 120 min following insulin injection as described above.

### 4.9. Western Blot

Equal amounts of protein extracted from the liver, muscle and adipose tissue homogenates were loaded and resolved on 4–15% SDS-polyacrylamide gels, and transferred to PVDF membrane (Millipore, Bedford, MA, USA). The membrane was blocked and probed with primary antibody (1:1000 dilution) for phospho-Akt (serine 473) (Cat#:4058), Akt (Cat#:4685), phospho-glycogen synthase kinase-3β (p-GSK-3β) (serine 9) (Cat#:5558), GSK-3β (Cat#:9315), and β-Actin (Cat#:4970) (Cell Signaling Technology, Inc., Danvers, MA, USA), overnight at 4 °C, and incubated with the corresponding horseradish peroxidase-conjugated secondary antibody. Protein signals were visualized using the enhanced chemiluminescence system (Amersham Biosciences, Little Chalfont, UK). Band intensities were quantified by the fold changes in terms of the ratios of phospho-proteins/total proteins using ImageJ software [http://rsb.info.nih.gov/ij/ (accessed on 28 December 2022)]. Ponceau S staining was used as the loading control.

### 4.10. Fatty Acid Gas Chromatography (GC) Analysis

Lipids were extracted from the rat liver (100 mg±) using the method of Folch–Lees [53]. The extracts were filtered, and lipids recovered in the chloroform phase. Individual lipid classes were separated by thin layer chromatography using Silica Gel 60 A plates developed in petroleum ether, ethyl ether, acetic acid (80:20:1) and visualized by rhodamine 6G. Phospholipids, diglycerides, triglycerides, free fatty acids, and cholesteryl esters were scraped from the plates and methylated using BF3/methanol as described by Morrison and Smith [54]. The methylated fatty acids were extracted and analyzed by gas chromatography. Gas chromatographic analyses were carried out on an Agilent 7890A gas chromatograph (Santa Clara, CA, USA) equipped with flame ionization detectors, a capillary column (SP2380, 0.25 mm × 30 m, 0.25 µm film, Supelco, Bellefonte, PA, USA). Helium was used as a carrier gas. The oven temperature was programmed from 160 °C to 230 °C at 4 °C/min. Fatty acid methyl esters were identified by comparing the retention times to those of known standards. The inclusion of lipid standards with odd chain fatty acids permitted quantitation of the amount of lipid in the sample. dipentadecanoyl phosphatidylcholine (C15:0), diheptadecanoin (C17:0), trieicosenoin (C20:1), and cholesteryl eicosenoate (C20:1) were used as standards.

### 4.11. Cholesterol GC Analysis

Unesterified Cholesterol: Internal standard was added to a portion of the lipid extract, concentrated under nitrogen and then solubilized in carbon disulfide to inject onto the gas chromatograph. Samples were analyzed on an Agilent 7890A gas chromatograph equipped with an HP-50+ column (0.25 mm i.d. × 30 m, 0.25 µm film, Agilent) and a flame ionization detector. The oven temperature was programmed from 260 °C to 280 °C and helium was used as the carrier gas [55].

### 4.12. Statistical Analysis

Data were expressed as mean ± SD (standard deviation) and analyzed using one-way ANOVA, followed by Tukey’s multiple comparison test or unpaired *t* test using GraphPad Prism 9.2.0. Differences were considered statistically significant at *p* < 0.05.

## Figures and Tables

**Figure 1 ijms-25-04836-f001:**
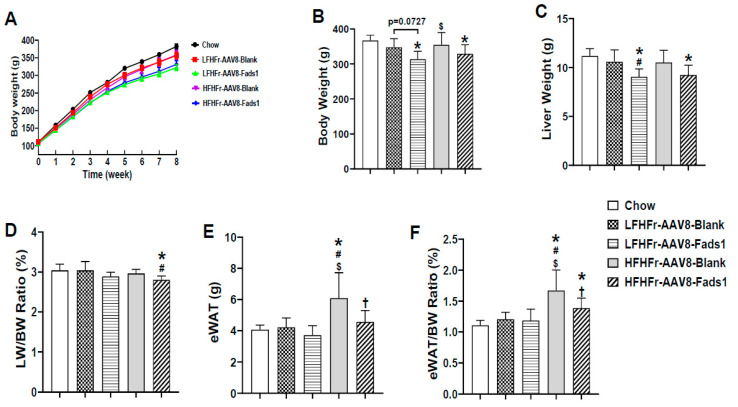
Effects of hepatocyte-specific *Fads1* overexpression on body weight, liver weight, and epididymal fat weight in rats fed with high fructose-containing diets. (**A**) Body weight (BW) dynamic change (nonfasted). (**B**) Body weight at end of experiment (fasted). (**C**) Liver weight (LW). (**D**) LW/BW ratio. (**E**) Epididymal fat weight (eWAT). (**F**) eWAT/BW ratio. Data represent means ± SD (n = 8). Statistical significance was set to *p* < 0.05, one-way ANOVA followed by Tukey’s multiple comparison test. * Versus chow; # versus LFHFr-AAV8-Blank; $ versus LFHFr-AAV8-Fads1; † versus HFHFr-AAV8-Blank. LFHFr, low-fat high-fructose diet; HFHFr, high-fat high-fructose diet.

**Figure 2 ijms-25-04836-f002:**
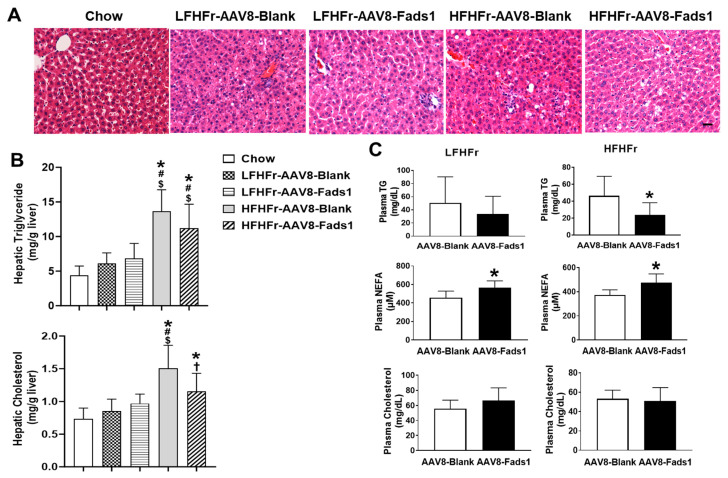
Effects of hepatocyte-specific *Fads1* overexpression on liver fat accumulation and plasma lipid profile. (**A**) Representative photomicrographs of the hematoxylin and eosin (H&E) staining of liver section (×200). Scale bar, 50 μm. (**B**) Hepatic triglyceride and cholesterol. (**C**) Plasma triglyceride, NEFA, and cholesterol. Data represent means ± SD (n = 8). Statistical significance was set to *p* < 0.05, one-way ANOVA followed by Tukey’s multiple comparison test (**B**) or unpaired *t* test (**C**). * Versus chow; # versus LFHFr-AAV8-Blank; $ versus LFHFr-AAV8-Fads1; † versus HFHFr-AAV8-Blank. LFHFr, low-fat high-fructose diet; HFHFr, high-fat high-fructose diet. NEFA, non-esterified fatty acid.

**Figure 3 ijms-25-04836-f003:**
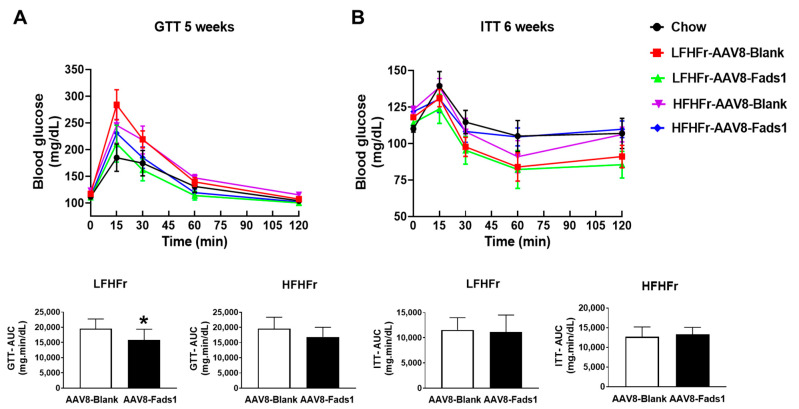
Effects of hepatocyte-specific *Fads1* overexpression on glucose and insulin tolerance tests. (**A**) Glucose tolerance test (GTT) and calculated areas under curve (AUC). (**B**) Insulin tolerance test (ITT) and calculated AUC. Data represent means ± SD (n = 8). * *p* < 0.05, unpaired *t* test. LFHFr, low-fat high-fructose diet; HFHFr, high-fat high-fructose diet.

**Figure 4 ijms-25-04836-f004:**
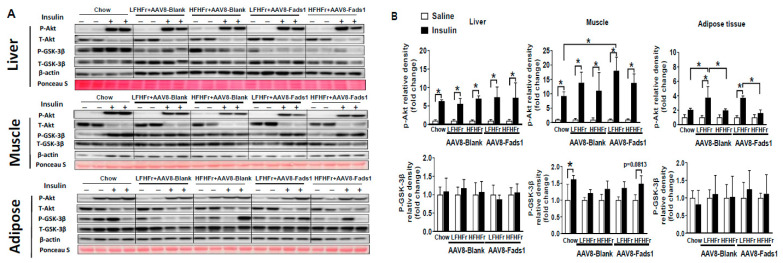
Effects of hepatocyte-specific *Fads1* overexpression on insulin signaling. After 8 weeks of high fructose-containing diet feeding, mice received a bolus injection of insulin (1 units/kg) through the portal vein. Five minutes later, tissues were harvested for Western blot analysis. Representative Western blot images (**A**) and quantification of protein expression (**B**) in the liver, muscle, and adipose tissue. Data represent means ± SD (n = 4). * *p* < 0.05, unpaired *t* test. LFHFr, low-fat high-fructose diet; HFHFr, high-fat high-fructose diet.

**Figure 5 ijms-25-04836-f005:**
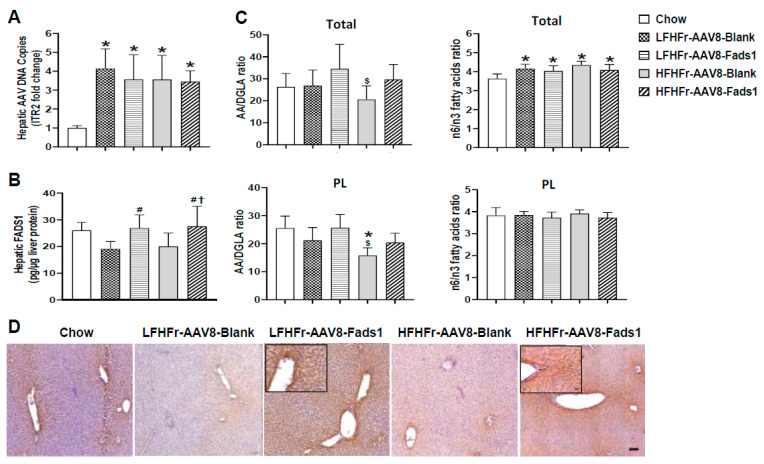
Hepatocyte-specific overexpression of *Fads1* increases hepatic FADS1 activity in LFHFr and HFHFr-fed rats. (**A**) ITR2 qPCR quantitation of AAV8 genome DNA copies in the liver. (**B**) Hepatic FADS1 assessed by ELISA. (**C**) AA/DGLA ratio and n6/n3 fatty acids ratio in total fatty acid and PL pools. (**D**) Representative photomicrographs of immunohistochemical staining of FADS1 in the liver section (×100). Scale bar, 100 μm; Insert (×400) scale bar, 20 μm. Data represent means ± SD (n = 8). Statistical significance was analyzed by one-way ANOVA followed by Tukey’s multiple comparison test. Statistical significance was set to *p* < 0.05. * Versus chow; # versus LFHFr-AAV8-Blank; $ versus LFHFr-AAV8-Fads1; † versus HFHFr-AAV8-Blank. ITRs, inverted terminal repeats; AA, arachidonic acid; DGLA, dihomo-gamma-linolenic acid; Pl, phospholipid; LFHFr, low-fat high-fructose diet; HFHFr, high-fat high-fructose diet.

**Figure 6 ijms-25-04836-f006:**
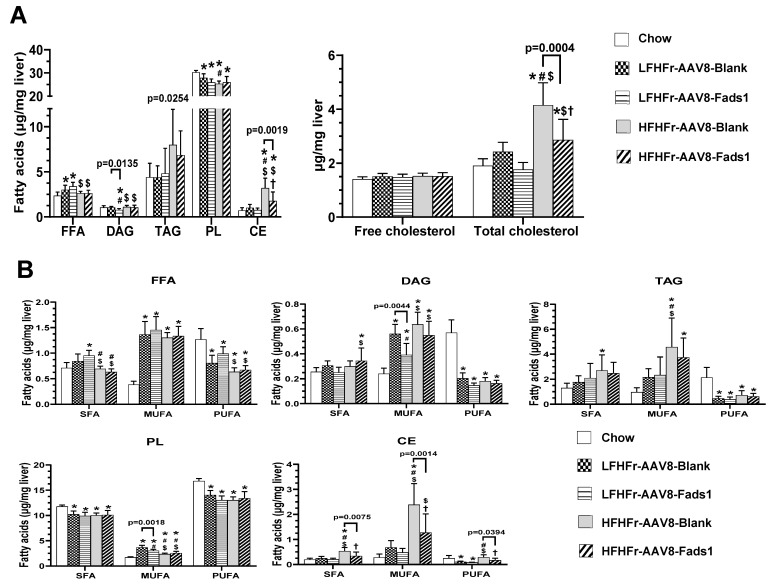
Effects of hepatocyte-specific overexpression of *Fads1* on the alteration of hepatic lipids in LFHFr and HFHFr-fed rats. (**A**) Hepatic total fatty acids by individual lipid compartment and cholesterol. (**B**) Hepatic total saturated fatty acids (SFA), monounsaturated fatty acids (MUFA), and polyunsaturated fatty acids (PUFA) by individual lipid classes. (**C**) Fatty acid composition by individual lipid classes. Data represent means ± SD (n = 8). Statistical significance was analyzed by one-way ANOVA followed by Tukey’s multiple comparison test. Statistical significance was set to *p* < 0.05. * Versus chow; # versus LFHFr-AAV8-Blank; $ versus LFHFr-AAV8-Fads1; † versus HFHFr-AAV8-Blank. PL, phospholipid; FFA, free fatty acid; DAG, diacylglycerol; TAG, triacylglycerol; CE, cholesteryl ester. LFHFr, low-fat high-fructose diet; HFHFr, high-fat high-fructose diet.

**Table 1 ijms-25-04836-t001:** Characterization of metabolic phenotypes.

Variable	Chow	LFHFr-AAV8-Blank	LFHFr-AAV8-Fads1	HFHFr-AAV8-Blank	HFHFr-AAV8-Fads1
Plasma ALT (U/L)	15.1 ± 2.7	8.1 ± 2.4 *	8.5 ± 2.2 *	10.1 ± 1.7 *	10.8 ± 1.6 *
Plasma AST (U/L)	35.2 ± 21.7	25.5 ± 17.7	22.3 ± 7.5	28.6 ± 16.2	18.7 ± 2.3
Cecum weight (CW, g)	4.94 ± 0.47	2.69 ± 0.42 *	2.40 ± 0.34 *	2.44 ± 0.37 *	2.51 ± 0.38 *
CW/BW (%)	1.35 ± 0.13	0.77 ± 0.12 *	0.77 ± 0.08 *	0.70 ± 0.13 *	0.76 ± 0.09 *
Total energy intake (Kcal/rat/day)	70.71 ± 8.48	61.97 ± 2.76 *	56.49 ± 1.82 *	65.75 ± 2.65 ^$^	65.10 ± 2.56 ^$^
Energy from fructose water (Kcal/rat/day)	0	18.76 ± 1.39 *	16.79 ± 1.04 *	19.55 ± 2.29 *^$^	19.78 ± 1.84 *^$^
Energy from pellet food (Kcal/rat/day)	70.71 ± 8.48	43.20 ± 1.8 *	39.70 ± 1.44 *	46.20 ± 1.28 *^$^	45.31 ± 1.62 *
Energy efficiency ratio (EER, %)	6.00 ± 0.36	5.42 ± 0.56	4.75 ± 0.42 *	5.50 ± 0.79	4.97 ± 0.58 *
Fasting blood glucose (mg/dL)	110 ± 19	139 ± 47	112 ± 31	133 ± 41	120 ± 41
Fasting blood insulin (ng/dL)	0.58 ± 0.07	0.54 ± 0.08	0.59 ± 0.06	0.61 ± 0.06	0.57 ± 0.08
HOMA-IR	3.90 ± 0.75	4.77 ± 2.29	4.03 ± 1.23	4.92 ± 1.29	4.25 ± 1.62

Statistical significance was set to *p* < 0.05, one-way ANOVA followed by Tukey’s multiple comparison test. * Versus chow; ^$^ versus LFHFr-AAV8-Fads1. LFHFr, low-fat high-fructose diet; HFHFr, high-fat high-fructose diet; HOMA-IR, homeostasis model assessment of insulin resistance.

## Data Availability

All data are contained within the manuscript.

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
