# Peer review of "Hepatocyte-Specific Fads1 Overexpression Attenuates Western Diet-Induced Metabolic Phenotypes in a Rat Model"

_ijms, 2024, doi:10.3390/ijms25094836_

Round 1
Reviewer 1 Report (Previous Reviewer 2)
Comments and Suggestions for Authors
The authors improved the manuscript with the observations mentioned above. The quality is adequate to be publishable,
Author Response
Response: We thank this reviewer for the agreement with our work.
Reviewer 2 Report (Previous Reviewer 1)
Comments and Suggestions for Authors
This manuscript (MS) written by Dushan T. Ghooray et al. is a revised version of previously rejected MS (ijms-2670605). Although the authors added a point-by-point response, their responses are limited and fail to respond all the comments I raised in the first round and second round of review. According to an added data (Figure 5B in this MS), expression levels of endogenous rFads1 in the liver were influenced by diet; the effect of AAV8-mediated induction of rFads1 on its protein levels was marginal, although the authors used an overexpression system. Indeed, rFads1 levels in the groups treated with AAV8-Fads1 were comparable those in the chow control group. Hence, the authors’ claims that the favorable phenotypic changes observed resulted from the hepatic overexpression of Fads1 are not convincing, which cannot allow the Reviewer to recommend the publication of this MS in International Journal of Molecular Sciences.
Author Response
Response: We thank this reviewer for the comments. We agree that “expression levels of endogenous rFads1 in the liver were influenced by diet”. Yes, that is true. LFHFr and HFHFr diets are refined diets and are unhealthy diets that were used to induce metabolic phenotypes. Our central hypothesis is that LFHFr and HFHFr diets lead to a lower level of hepatic Fads1, which is a likely causal factor leading to the development of NAFLD compared to the chow diet. This is consistent with human studies that reduced Fads1 was observed in NAFLD/NASH patients (PMID: 28436449, 25123259), which relates to our central premise. Although the effect of AAV8-mediated induction of rFads1 on its protein levels was marginal, the difference of Fads1 level was statistically significant. The “rFads1 levels in the groups treated with AAV8-Fads1 were comparable with those in the chow control group” is in line with our hypothesis, that means the Fads1 level in the rats treated with AAV8-Fads1 was restored to the healthy control level (chow fed rats). We expect that multiple repeated injection of AAV8-Fads1 will have a better effect than the single injection. However, we did observe that some metabolic phenotypes were improved, even with a single injection of AAV8-Fads1.
We hope this convinces the reviewer.
Reviewer 3 Report (New Reviewer)
Comments and Suggestions for Authors
Abstract:
· Results: Quantify significant findings to underscore the study's impact.
· Significance: Briefly emphasize the novel contribution and potential implications for NAFLD research.
1. Introduction
· In general, the text is ordered and clear. To improve readability, the sentence structure may occasionally be more succinct. For easier understanding, the explanation of the different fatty acids and their metabolic processes might be condensed.
· The introduction provides a comprehensive overview of the state of the field by discussing several papers and including the necessary citations. To strengthen the case for the study's importance, a more critical examination of contradictory conclusions or limits in the body of current research would be beneficial.
Suggestions for Improvement:
· Make difficult phrases simpler and make sure technical jargon are clarified or pertinent to the study topic to improve intelligibility.
· While the introduction provides a good overview of Fads1 and its potential implications in NAFLD, a more explicit connection between this background and the study's objectives could strengthen the narrative.
· Consider a brief discussion on the potential translational impact or clinical relevance of the findings to underscore the significance of the research.
2. Results
· The results are presented in a thorough and organized manner, with distinct subheadings that help the reader navigate the information. The correct referencing of figures and tables facilitates the data's display and interpretation. On the other hand, maintaining terminology uniformity and steering clear of recurring expressions would improve lucidity.
· The authors evaluate their data using acceptable statistical techniques, which increases the validity of their results. It is imperative that all statistical tests, p-values, and sample sizes be specified explicitly since they play a critical role in determining how reliable the results are.
· The authors show a relationship between the genetic alteration and the phenotypic outcomes by attributing the observed metabolic benefits to hepatocyte-specific Fads1 overexpression. More research would be helpful in determining if these alterations are a direct consequence of overexpressing Fads1 or whether they are indirect impacts.
· While the results add valuable insights into the role of Fads1 in metabolic regulation, placing these findings within the broader context of existing research would provide a clearer understanding of their novelty and significance. Discussing how these results compare with or differ from previous studies could enhance the manuscript's impact.
3. Discussion
· The discussion connects the study's findings well to the larger context of metabolic and NAFLD studies. It describes how hepatocyte-specific Fads1 overexpression affects metabolic characteristics, indicating a thorough comprehension of the study's consequences. You may improve this section by presenting a more thorough comparison with other pertinent research, emphasizing findings that are in agreement and disagreement.
· The explanation of how hepatic lipid metabolism and insulin signaling may be impacted by Fads1 overexpression is informative. Though possibly by including more current research or putting forward original theories in light of your results, the section might use a more thorough examination of these systems.
· While speculative insights can be valuable, ensure they are well-grounded in existing literature or the study's data. When proposing future research directions or hypothesizing about mechanisms, clearly delineate these as speculative to avoid overinterpretation.
· Discussing the potential therapeutic implications of your findings is an essential aspect of this section. Emphasizing how these findings might translate into clinical strategies or interventions for NAFLD could enhance the discussion's relevance and impact.
· Ensure that your discussion is well-integrated with the data presented in the results section, referencing specific figures and tables where relevant to strengthen your arguments.
4. Materials and Methods
· Section 4.1: All animal handling, experimental design, and treatment methods are described in detail and in accordance with the relevant ethical guidelines. The inclusion of the study's institutional committee approval, which emphasizes adherence to ethical norms, is commendable. To ensure that the study can be replicated, it would be helpful to include further information on the diet composition and the reasoning behind the animal model selection.
· Section 4.2: The assays are suited for the study and standard; they are used to test various parameters. It would be better to make this section clearer by stating how certain kits were chosen and any changes made to conventional protocols.
· Section 4.3: Detailed information about the primers and PCR conditions is provided in this part. Including the transduction effectiveness (i.e., the proportion of transduced cells) might offer further information about the experiment's outcome.
· Section 4.4: A sufficient description of the ELISA technique for FADS1 detection is provided. Information on the assay's sensitivity, specificity, and any detected cross-reactivity would be helpful.
· Section 4.5: The protocols for tissue preparation and staining are well-defined and standardized. This section would be strengthened by the antibody validation (specificity, titration).
· Section 4.6: The technique provided is a common way to measure hepatic triglycerides. It might be improved by describing any possible interference seen, as well as the method's sensitivity and specificity.
· Sections 4.7 and 4.8: These sections are understandable and include enough information on replication. Giving the justification for the selected insulin and glucose dosages might provide further understanding of the experimental setup.
· Section 4.9: This subsection, which covers the circumstances and sources of antibodies, is sufficiently thorough. However, the data' repeatability and clarity would be enhanced by providing information on the quantification process and antibody specificity confirmation.
· Sections 4.10 and 4.11 provide a thorough description of the gas chromatography techniques used to analyze cholesterol and fatty acids. This part would be improved with further details on calibration, standard curves, and technique validation.
· Section 4.12: The statistical techniques have been carefully selected and explained. Gaining a comprehensive understanding of the data analysis would need clarifying the tests selected for particular data kinds and verifying any presumptions.
Comments on the Quality of English Language
While the text is generally understandable, several instances of grammatical errors, awkward phrasing, and inconsistencies in terminology are apparent. These issues could hinder the manuscript's clarity and detract from its professional presentation. Moderate editing would enhance the readability and overall quality of the manuscript.
Author Response
Response: We thank this reviewer for the helpful and detailed comments. We will address the concerns point-to-point.
Abstract:
- Results: Quantify significant findings to underscore the study's impact.
- Significance: Briefly emphasize the novel contribution and potential implications for NAFLD research.
- Introduction
- In general, the text is ordered and clear. To improve readability, the sentence structure may occasionally be more succinct. For easier understanding, the explanation of the different fatty acids and their metabolic processes might be condensed.
- The introduction provides a comprehensive overview of the state of the field by discussing several papers and including the necessary citations. To strengthen the case for the study's importance, a more critical examination of contradictory conclusions or limits in the body of current research would be beneficial.
Suggestions for Improvement:
- Make difficult phrases simpler and make sure technical jargon are clarified or pertinent to the study topic to improve intelligibility.
- While the introduction provides a good overview of Fads1 and its potential implications in NAFLD, a more explicit connection between this background and the study's objectives could strengthen the narrative.
- Consider a brief discussion on the potential translational impact or clinical relevance of the findings to underscore the significance of the research.
Response: We thank this reviewer for the kind suggestions.
We have made efforts to improve the abstract and introduction in several lines.
- Polished the abstract by quantifying the significant findings and emphasizing the novel contribution of the study.
- Shortened the discussion of LCPUFAs metabolic processes to make the introduction section more succinct.
- Rephrased some of the long sentences.
- Reorganized the logic flow to make the connections between paragraphs more streamlined.
- Examined recent studies with contradictory conclusions and justified our study.
- Results
- The results are presented in a thorough and organized manner, with distinct subheadings that help the reader navigate the information. The correct referencing of figures and tables facilitates the data's display and interpretation. On the other hand, maintaining terminology uniformity and steering clear of recurring expressions would improve lucidity.
- The authors evaluate their data using acceptable statistical techniques, which increases the validity of their results. It is imperative that all statistical tests, p-values, and sample sizes be specified explicitly since they play a critical role in determining how reliable the results are.
- The authors show a relationship between the genetic alteration and the phenotypic outcomes by attributing the observed metabolic benefits to hepatocyte-specific Fads1 overexpression. More research would be helpful in determining if these alterations are a direct consequence of overexpressing Fads1 or whether they are indirect impacts.
- While the results add valuable insights into the role of Fads1 in metabolic regulation, placing these findings within the broader context of existing research would provide a clearer understanding of their novelty and significance. Discussing how these results compare with or differ from previous studies could enhance the manuscript's impact.
Response: We agree with the reviewer. We have made the terminology uniform in the entire text and statistical tests specified. We agree that more research would be helpful to investigate the mechanisms underlying the beneficial effects of Fads1 overexpression. The increased LCPUFA along with reduced MUFAs by AAV8-Fads1 suggests that Fads1 overexpression may inhibit lipogenesis (re-esterification) through FADS1 derived LCPUFA. This will be our future direction. We also discussed the difference between our study and previous ones (in the discussion section).
- Discussion
- The discussion connects the study's findings well to the larger context of metabolic and NAFLD studies. It describes how hepatocyte-specific Fads1 overexpression affects metabolic characteristics, indicating a thorough comprehension of the study's consequences. You may improve this section by presenting a more thorough comparison with other pertinent research, emphasizing findings that are in agreement and disagreement.
- The explanation of how hepatic lipid metabolism and insulin signaling may be impacted by Fads1 overexpression is informative. Though possibly by including more current research or putting forward original theories in light of your results, the section might use a more thorough examination of these systems.
- While speculative insights can be valuable, ensure they are well-grounded in existing literature or the study's data. When proposing future research directions or hypothesizing about mechanisms, clearly delineate these as speculative to avoid overinterpretation.
- Discussing the potential therapeutic implications of your findings is an essential aspect of this section. Emphasizing how these findings might translate into clinical strategies or interventions for NAFLD could enhance the discussion's relevance and impact.
- Ensure that your discussion is well-integrated with the data presented in the results section, referencing specific figures and tables where relevant to strengthen your arguments.
Response: We agree with the reviewer. We incorporated all the suggestions by this reviewer in the discussion. We compared our study with other relevant studies and emphasized the differences. We also added the figures related to our findings in the discussion and clearly delineated our suppositions. Moreover, we pointed out the strategy for future translational research. In addition, we condensed the discussion and made the terminology uniform.
- Materials and Methods
- Section 4.1: All animal handling, experimental design, and treatment methods are described in detail and in accordance with the relevant ethical guidelines. The inclusion of the study's institutional committee approval, which emphasizes adherence to ethical norms, is commendable. To ensure that the study can be replicated, it would be helpful to include further information on the diet composition and the reasoning behind the animal model selection.
Response: We have added further information on diet composition, and more detailed information can be found with the diet catalog number. The reason that we chose the SD rats is based on our previous study which has been referenced.
- Section 4.2: The assays are suited for the study and standard; they are used to test various parameters. It would be better to make this section clearer by stating how certain kits were chosen and any changes made to conventional protocols.
Response: We chose the kits based on our previous study (has been referenced) and exactly followed the manufacturer’s instruction.
- Section 4.3: Detailed information about the primers and PCR conditions is provided in this part. Including the transduction effectiveness (i.e., the proportion of transduced cells) might offer further information about the experiment's outcome.
Response: The hepatic expression of AAV specific sequences (ITR2) were comparable between the groups that received AAV8 injection and was robustly increased compared to non-transfected rats, suggesting transduction effectiveness.
- Section 4.4: A sufficient description of the ELISA technique for FADS1 detection is provided. Information on the assay's sensitivity, specificity, and any detected cross-reactivity would be helpful.
Response: We have added more information on the assay’s sensitivity, specificity and cross-reactivity.
- Section 4.5: The protocols for tissue preparation and staining are well-defined and standardized. This section would be strengthened by the antibody validation (specificity, titration).
Response: The negative control (PBS without primary antibody) was performed to validate the specificity of antibody and 1:100 dilution was used as recommended by the manufacturer.
- Section 4.6: The technique provided is a common way to measure hepatic triglycerides. It might be improved by describing any possible interference seen, as well as the method's sensitivity and specificity.
Response: This method is well established and reliable, and it has been used by numerous investigators in the area of MASLD research.
- Sections 4.7 and 4.8: These sections are understandable and include enough information on replication. Giving the justification for the selected insulin and glucose dosages might provide further understanding of the experimental setup.
Response: These dosages were selected based on our previous work and published paper (referenced).
- Section 4.9: This subsection, which covers the circumstances and sources of antibodies, is sufficiently thorough. However, the data' repeatability and clarity would be enhanced by providing information on the quantification process and antibody specificity confirmation.
Response: The band density was quantified by the fold changes in terms of the ratios of phospho-proteins/total proteins. All antibodies in the WB study are monoclonal antibodies with high specificity and the detailed antibody information can be found with the catalog number.
- Sections 4.10 and 4.11 provide a thorough description of the gas chromatography techniques used to analyze cholesterol and fatty acids. This part would be improved with further details on calibration, standard curves, and technique validation.
Response: The quantification of fatty acids and cholesterol was determined by adding known amounts of standards. Relative peak areas were measured as retention distance × peak height and weight percentage compositions calculated by applying correction factors (PMID: 14221106). This is a well-established method and was analyzed in Vanderbilt University Medical Center (VUMC) lipid core, which is a designated service core by NIH MMPC.
- Section 4.12: The statistical techniques have been carefully selected and explained. Gaining a comprehensive understanding of the data analysis would need clarifying the tests selected for particular data kinds and verifying any presumptions.
Response: We agree and have clarified the tests in each dataset (see figure legends).
Comments on the Quality of English Language
While the text is generally understandable, several instances of grammatical errors, awkward phrasing, and inconsistencies in terminology are apparent. These issues could hinder the manuscript's clarity and detract from its professional presentation. Moderate editing would enhance the readability and overall quality of the manuscript.
Response: English language editing has been done by our colleague-Marion McClain, who is a native English speaker and is fluent with English writing and editing.
Round 2
Reviewer 2 Report (Previous Reviewer 1)
Comments and Suggestions for Authors
This manuscript (MS) written by Dushan T. Ghooray et al. is a revised version of the revised version of previously rejected MS (ijms-2670605). Although the authors made a rebuttal comment, their responses have been limited and fail to respond all the comments I raised in previous three rounds of review. Indeed, there is no point-by-point response in this MS during all rounds of peer-reviews. Moreover, all figures are not presented in the uploaded PDF for the Reviewer unfortunately; nonetheless, considering the flow of the authors’ comment and the revised MS, the Reviewer guess that no data must have been added in this round of review. Given the insufficiency in the scientific soundness of this study including the logic flow of MS, the Reviewer cannot recommend the publication of this MS in International Journal of Molecular Sciences.
This manuscript is a resubmission of an earlier submission. The following is a list of the peer review reports and author responses from that submission.
Round 1
Reviewer 1 Report
Comments and Suggestions for Authors
This manuscript (MS) written by Dushan T. Ghooray et al. investigated the effects of rat fatty acid desaturase 1 (Fads1) overexpression in the liver, mediated by a tissue specific manner using an adenovirus-associated virus (AAV) vector containing an albumin promoter, on metabolic phenotypes of rat fed with western [low-fat high-fructose, LFHF; high-fat high-fructose, HFHF] diets. The authors observed a positive effect they expected in some parameters; however, the beneficial effects were seemed to be limited in the data, which cannot fully support the authors’ claim that their study provides a proof-of-concept that hepatocyte FADS1 activation is beneficial in the improvement of NAFLD and associated metabolic phenotypes. Given some concerns described below, data shown in the MS should be carefully interpreted, which cannot allow the Reviewer to recommend the publication of this MS in International Journal of Molecular Sciences at the current form. Also, the Reviewer reserves other comments at this moment.
Major comments:
In this study, the authors examined the arachidonic acid (AA)/Dihomo-γ-linolenic acid (DGLA) ratio as a surrogate marker of delta-5 desaturase activity (function of Fads1). However, this activity has not always a linear relationship with the expression levels of Fads1 as shown in a knockout study (PMID: 22534642) and AA and DGLS levels also depend on the hepatic levels of other fatty acids and other enzyme activities in a metabolic context, theoretically. Thus, the successful achievement of rFads1 induction in this study should be directly demonstrated and quantitatively evaluated. Also, expression levels of endogenous rFads1, which may have been affected by diet according to the data, should be carefully handled. Moreover, the AAV construction for rFads1 (functional expression of rFad1 and its hepatic specificity) was not validated in experimental conditions the authors employed.
In some Figures, data of chow control group are not shown.
Minor comments:
In Figure 2, hepatic levels of each FA should be demonstrated in addition to the ratios.
Statistic significant should be < 0.05, not ≤ 0.05. Also, a multiple comparison should be conducted when the sample groups are more than three.
Immunoblot images have no internal controls; sectional images have no scale bars.
For potential readers, it will be benefitial to show control group and corresponding sample group as next to each other as possible as. Also, bar colors should be unified through the MS, especially in Figure 6.
Reviewer 2 Report
Comments and Suggestions for Authors
The article is interesting and mentions a relevant topic such as the possible treatment for Metabolic Syndrome. It is always interesting to see results with high-calorie diets that are studied in short periods of time to see the short-term effects. However, there is an important problem in the design of the study that would cause the results to not be analyzed in the best way. The lack of a CHow control group with via the tail vein injection of recombinant AAV8 thus creates important doubts, why do the LFHFr-AAV8-Blank groups and especially the HFHFr-AAV8-Blank group weigh less than the Chow? Saying AAV8-Blank, I suppose it lacks the virus injection and therefore those fed a high-alority diet should at least weigh the same. This can be seen especially in the weight results, as the Chow group weighed the most when those on a high-calorie diet should logically weigh more than them.
Were any vehicles injected into the AAV8-Blank groups? Were the animals stressed in some way to eat less? Was the amount of food ingested by the animals analyzed? Was the data on the amount of water and fructose consumed per day analyzed? These could be reasons why the animals, instead of gaining weight, lost weight, so the results could have some bias in their analysis.
It should be noted that both the blots and the images of the tissues seem to be quite well done, so this is a great merit of the authors. However, the previously mentioned doubts must be resolved in order to understand the article.
Comments on the Quality of English Language
The article is interesting and mentions a relevant topic such as the possible treatment for Metabolic Syndrome. It is always interesting to see results with high-calorie diets that are studied in short periods of time to see the short-term effects. However, there is an important problem in the design of the study that would cause the results to not be analyzed in the best way. The lack of a CHow control group with via the tail vein injection of recombinant AAV8 thus creates important doubts, why do the LFHFr-AAV8-Blank groups and especially the HFHFr-AAV8-Blank group weigh less than the Chow? Saying AAV8-Blank, I suppose it lacks the virus injection and therefore those fed a high-alority diet should at least weigh the same. This can be seen especially in the weight results, as the Chow group weighed the most when those on a high-calorie diet should logically weigh more than them.
Were any vehicles injected into the AAV8-Blank groups? Were the animals stressed in some way to eat less? Was the amount of food ingested by the animals analyzed? Was the data on the amount of water and fructose consumed per day analyzed? These could be reasons why the animals, instead of gaining weight, lost weight, so the results could have some bias in their analysis.
It should be noted that both the blots and the images of the tissues seem to be quite well done, so this is a great merit of the authors. However, the previously mentioned doubts must be resolved in order to understand the article.
Round 2
Reviewer 1 Report
Comments and Suggestions for Authors
The authors made a revision to improve the original manuscript. Nonetheless, unfortunately, additional experiments were not conducted against my comments. The reviewer agree with that AAV8-mediated overexpression in the liver is a well-documented method; however, this does not mean that the authors successfully conducted it. Importantly, immunohistochemical staining images presented cannot be quantified, given its methodological theory and differences in background among the data shown in the manuscript. In each experimental condition, the authors should quantitatively evaluate the effects of transduced Fads1 (which include infection efficiency) apart from those of endogenous Fads1.
Regarding the relationship of AA/DGLA ratio with the expression levels of Fads1, the authors should experimentally make a proof in their experimental system.
For immunoblotting, some data of β-actin were added. But they are not fine data for the liver and muscle, which needs re-experiment.
Reviewer 2 Report
Comments and Suggestions for Authors
The authors made some changes to the manuscript, nevertheless, the additional information rqueired on my last comments were not conducted. The reviewer agree with that the virus injection may play a role in body weight; however, the authors did not include the information that is crucial for the comparation with all the groups. The authors did recognize that the end body weight of chow group is higher than the other groups (even of the hypercaloric ones). Thus, what is the use of that control group? In the writing it is not understood whether or not there is a control group fed with chow and with the vector even without the Fads1 vector. If the wording of this part is not clear then the study loses all meaning since the analysis is impossible.
Comments on the Quality of English Language
Some sentences need to be redrafted.